# Exploring high mortality rates among people with multiple and complex needs: a qualitative study using peer research methods

Rachel Perry,[1] Emma A. Adams [ID],[1] Jill Harland,[1] Angela Broadbridge,[2] Emma L. Giles [ID],[3] Grant J. McGeechan [ID],[4] Amy O'Donnell [ID],[1] Sheena E. Ramsay[1]

¹Population Health Sciences Institute, Newcastle University, Newcastle upon Tyne, UK
²Fulfilling Lives Newcastle Gateshead, Gateshead, UK
³School of Health and Life Sciences, Teesside University, Middlesbrough, UK
⁴Centre for Applied Psychological Science, Teesside University, Middlesbrough, UK

**Correspondence to**
Emma A. Adams;
emma.adams@newcastle.ac.uk

## ABSTRACT

**Objective** To explore the perceived reasons underlying high mortality rates among people with multiple and complex needs.

**Design** Qualitative study using peer research.

**Setting** North East of England.

**Participants** Three focus group discussions were held involving (1) people with lived experience of multiple and complex needs (n=5); (2) front-line staff from health, social care and voluntary organisations that support multiple and complex needs groups (n=7); and (3) managers and commissioners of these organisations (n=9).

**Results** Findings from this study provide valuable perspectives of people with multiple complex needs and those that provide them with support on what may be perceived factors underlying premature mortality. Mental ill health and substance misuse (often co-occurring dual diagnosis) were perceived as influencing premature mortality among multiple and complex needs groups. Perceptions of opportunities to identify people at risk included critical life events (eg, bereavement, relationship breakdown) and transitions (eg, release from prison, completion of drug treatment). Early prevention, particularly supporting young people experiencing adverse childhood experiences, was also highlighted as a priority.

**Conclusion** High mortality in multiple and complex needs groups may be reduced by addressing dual diagnosis, providing more support at critical life events and investing in early prevention efforts. Future interventions could take into consideration the intricate nature of multiple and complex needs and improve service access and navigation.

## Strengths and limitations of this study

► This study employed focus group discussions with individuals with multiple and complex needs and service providers to explore factors perceived to contribute to high premature mortality rates in individuals experiencing multiple and complex needs.

► Peer researchers contributed to all stages of this study, including developing the aims, data collection and interpretation, and shaping recommendations.

► Using peer researchers enhanced our access to participants and improved interpretation of data.

► The main limitation is that the study only recruited individuals in one region in the North East of England. Views from individuals with multiple and complex needs and service providers in other areas of England might have led the results to being more generalisable; however, this study was focused at the local level.

of England.[2 3] This region has also observed high rates of drug-related deaths and suicide in younger adults (under age 50) compared with other parts of the UK.[4]

People facing co-occurring issues of homelessness, substance misuse, offending and mental ill health experience particularly poor health outcomes compared with the general population.[5] Multiple complex needs (MCN) has been used to describe this group; other terminology include severe and multiple disadvantage or multiple exclusion homelessness.[5–8] Homeless individuals have greater prevalence of alcohol use disorders than the general population,[9] and self-reported data suggest that mental health problems are as high as 92% among individuals with MCN.[5] When considered in isolation, homelessness, substance misuse, mental ill health and offending each contributes to higher mortality rates compared with the general

## INTRODUCTION

Despite increases in life expectancy in the UK throughout the 20th century, improvements in mortality rates have stalled in recent years for most population groups.[1] Since 2010–2011, the gap in life expectancy between people living in the most and least deprived communities has widened, with the largest decreases seen in the 10% most deprived neighbourhoods in the North East

population.[10–13] However, there is a growing appreciation of the intersecting nature of these needs and the compounding effect these can have on health outcomes.[5] The connection between homelessness and premature mortality has been highlighted through a number of quantitative studies.[4 10 13–17] Epidemiological data suggest that the contribution of mental ill health and co-occurring substance misuse to mortality rates is high among people experiencing MCN.[13 17] All-cause mortality rates for people with MCN compared with the UK general population are almost seven times higher for men and almost twelve times higher for women, with major causes of deaths including injury, poisoning, external causes (ie, accidents, homicide or suicide) and social exclusion.[14] However, most of the research into mortality within this population has been epidemiological in nature and there is a gap in qualitative research.

Despite the breadth of quantitative and epidemiological data on MCN and mortality, there are very limited perspectives from individuals with lived experienced of MCN and those who support them (eg, health and social care workers) on the high death rates in this population. A recent mapping exercise of MCN across England found the current system of support across public and voluntary services was struggling to meet present demands and deliver positive outcomes for individuals with MCN.[5] This is increasingly problematic when the system is fragmented and siloed, causing added barriers to much needed support.[18 19] With a system that is not designed to meet the needs of MCN populations, there is a pressing need to conduct research to explore the perspectives of both service users and providers to develop a more comprehensive picture of the current challenges and potential opportunities for improvements.

Involving peer researchers across the continuum of research, from concept development to data collection, analysis and dissemination, can help provide valuable access to participants from marginalised populations, as well as offer unique insights on how evidence should be collected and interpreted.[8 20–22] Peer researchers are individuals who have lived experience of the phenomenon under study and are at a stage where they are ready to support others discuss their own experiences.[8] Conducting research *on* MCN populations can be extremely challenging; however, conducting studies *with* individuals with lived experience of MCN has been found to add immense value through enhancing recruitment efforts and the collective understanding of the experiences.[23–25] Although there is increasing recognition of the need to involve peer researchers in health and social care research, there remains limited representation in current literature on mortality and MCN.

The aim of this study was to explore the perspectives of people with lived experience of MCN and professionals who support them in order to understand the factors perceived to contribute to high mortality rates in this population. This study uses peer research methods to obtain perspectives on premature mortality and rich insights into opportunities for preventing deaths and improving service provision for MCN groups in the future.

## METHODS

A qualitative study using a peer research approach was conducted. Peer researchers from Fulfilling Lives Newcastle Gateshead, a project that supports MCN groups, contributed to all stages of this study, including developing the aims, data collection and interpretation, and shaping recommendations. Peer researchers received National Vocational Qualification-level training in peer research and had lived experience of MCN. Peer researchers identified the study topic as an area that was personally impacting them and required attention. Additionally, peer researchers determined the data collection approach (focus group discussions, FGDs), co-developed the topic guide, assisted in facilitating focus groups, contextualised the findings, selected the quotes, presented the findings at a dissemination event and coauthored a blog on the topic.

### Data collection

The study sample comprised people with lived experience of MCN, front-line staff and managers/commissioners of relevant health, social care and voluntary support services. Peer researchers decided to use FGDs to help create a supportive environment for conversations around a sensitive topic. A combination of convenience (using pre-existing networks) and maximum variation sampling techniques was used to recruit participants through email and mailing lists within the North East of England for three homogenous FGDs. Participants were recruited from both operational (front-line) and commissioning/managerial levels to ensure diversity of staff perspectives. Additionally, variation was sought in terms of gender, different components of MCN and age of participants with lived experience, although all were aged 18 and over. Many peer researchers had relationships with FGD participants who have lived experience; however, this was a deliberate strategy as it enhanced our access to participants and improved the interpretation of data.

A participant information sheet was made available to all participants and informed consent was gained prior to data collection.

Three homogenous FGDs were conducted in June/July 2019 with individuals with lived experience of MCN (FGD 1, n=5), front-line staff (FGD 2, n=7) and managers/commissioners (FGD 3, n=9) working in health, social care and voluntary sectors. Representatives from a range of voluntary and statutory organisations participated, including those working in local authority commissioning, mental health, substance misuse, and housing and family services. Available participant characteristics are reported in table 1. No participants withdrew during the FGDs.

**Table 1** Characteristics of participants from the three focus groups

| Individuals with lived experience FGD | Front-line staff FGD | Manager/commissioner FGD |
|---|---|---|
| 5 participants. | 7 participants. | 9 participants. |
| ► 2 female. | ► 6 female. | ► 6 female. |
| ► 3 male. | ► 1 male. | ► 3 male. |
| | 6 different organisations representing service provision for the following: | 8 different organisations representing service provision for the following: |
| | ► 2 mental health. | ► 2 housing. |
| | ► 1 housing. | ► 2 mental health. |
| | ► 1 homelessness. | ► 2 MCN. |
| | ► 1 MCN. | ► 1 local council/government. |
| | ► 1 drug and alcohol. | ► 1 drug and alcohol. |

FGD, focus group discussion; MCN, multiple complex needs.

The FGDs were jointly facilitated by an academic and a peer researcher and held at accessible venues in Newcastle upon Tyne. Two semistructured topic guides (table 2) were co-produced with peer researchers, one for use with individuals with MCN and one for staff FGDs, with probes used to elicit additional information and detailed responses. All participants were fully briefed on the study (aims, objectives and dissemination plans) prior to gaining consent. FGDs lasted approximately 90 min and were audio-recorded.

### Data analysis

Audio recordings were transcribed verbatim and uploaded into NVivo V.12 to support data management and analysis. To protect anonymity, particularly given the sensitivity of the topic, any potential identifiers (including participant characteristics such as age, gender and role) were removed. Field notes were used to capture immediate reflections on the FGDs and to aid analysis and interpretation. Peer researchers reviewed transcripts for authenticity. Deductive thematic analysis based on Braun and Clarke's[26] approach was conducted. Initial codes were co-created with peer researchers based on the study aim and objectives, existing literature and an initial review of the transcripts. Each transcript was coded while adaptively developing the code book with

any new codes and returning to previous transcripts. The first author conducted the primary coding and a section of each transcript was second coded by two other authors to ensure consistency. Codes were collated into potential themes, which were reviewed among the research team and further refined. In addition to peer researchers, the team comprised academic researchers, service providers and public health registrar trainees with prior research experience. Consensus was reached through discussions with peer researchers and the wider research team on the themes, subthemes, extracts and recommendations.

### Patient and public involvement statement

At the time of this study, one of the coauthors (AB) worked for Fulfilling Lives Newcastle Gateshead, which is an 8-year learning programme that aims to improve the lives of people with MCN and build a trauma-informed approach with the services that support them. The project design and methodology were developed with a group of experts by experience (peer researchers) and front-line staff who work for Fulfilling Lives Newcastle Gateshead. As previously mentioned, peer researchers contributed to all stages of the research. The methodology selected for data collection (FGD) was chosen based on suggestions from peer researchers.

**Table 2** Topic guides for the focus groups

| Topic guide for focus groups with individuals with lived experience of MCN | Topic guide for focus groups with staff working with MCN groups |
|---|---|
| ► Awareness of mortality within their peer group. | ► Awareness of premature mortality within multiple and complex needs groups. |
| ► What factors/life experiences do they think contribute to premature mortality within their peer group. | ► Awareness of risk factors for premature mortality. |
| ► Any concerns they have about this personally or for others. | ► Current approaches to identify those at risk—perceptions of effectiveness. |
| ► Do they think anything could have been done to prevent people from dying? | ► What would help the services identify/target those at risk. |
| ► Can they describe this? | ► Current interventions—perceptions of effectiveness. |
| ► What types of help and support would they like to see being developed/provided? | ► Types of interventions/approaches they think should be in place. |
| ► How would this be best offered? | ► How could this be taken forward. |

MCN, multiple complex needs.

## RESULTS

Findings were relatively homogenous across all three FGDs. Findings are presented according to the two main study objectives and associated themes: *understanding factors underlying premature mortality* among MCN groups and *opportunities for interventions to prevent mortality*. Quotes most in agreement are presented in tables 3 and 4. Quotes from individuals with lived experience of MCN were more likely to focus on their personal experience as opposed to system factors as identified by staff groups. There was a high level of agreement in data collection both within and across the focus groups.

### Understanding factors underlying premature mortality in MCN groups
#### Burden of mental ill health and substance misuse issues
The severe burden of mental ill health in people with MCN was a recurrent concern and was felt to make a substantial contribution to premature mortality, particularly in relation to mental illness alongside substance misuse (dual diagnosis). FGD participants with lived experience shared experiences and perceptions of drugs being used as a way for individuals to escape from their day-to-day circumstances.

Across all three FGDs, there was a perception that people with MCN experienced challenges when trying to access services and support for coexisting mental ill health and substance misuse. Services were set up to deal with either mental health issues or addiction issues, meaning those with a dual diagnosis failed to receive appropriate support.

#### Lack of hope, stigma and health-seeking behaviour
Participants with MCN highlighted that experiences of loss were so common that it led to individuals becoming desensitised to deaths and reaching a point of acceptance. Outcomes for individuals experiencing MCN were so often negative this led to a lack of hope that it could ever be any different.

Stigma was discussed among the FGD with individuals with lived experience in relation to the large role it was perceived to play in health-seeking behaviour. Individuals with lived experience expressed frustration around the facial expressions and stigmatising nature of the words professionals used to describe different facets of MCN. This fear of stigma along with not being given sufficient time for staff to understand their unique experiences led to individuals avoiding services. FGDs with staff reiterated this perception that many young people they worked with felt intimidated when going to see their general practitioner and instead chose to self-medicate with illicit substances.

#### Poor navigation and limited services
Across all three FGDs, participants expressed problems with navigating service pathways, limited services and having to engage with traditional models of care. Individuals with MCN felt as though they were faced with a series of barriers to gaining appropriate support, which increased the vulnerability of MCN groups to premature mortality. There was a perception that recent reductions in available support services meant that the support was often only provided in extreme circumstances, such as when an individual had reached a crisis point.

### Opportunities for interventions to prevent mortality in MCN groups
#### Intervention timing
Critical life events (eg, childhood adversity and bereavement) and transitions (eg, release from prison) were highlighted as moments of key vulnerability, as well as potential opportunities for intervention. Participants felt that many of the issues encountered by people experiencing MCN were rooted in early childhood. Therefore, improved support for young people who experience adverse childhood experiences could prevent the development and exacerbation of long-term needs. Participants in all focus groups agreed that windows of opportunity (time points of places where services could be provided) are often brief and difficult to take advantage of.

#### Intervention approaches
Across all FGDs, participants provided suggestions for how interventions could be improved in the future. The need for a holistic, person-centred approach was highlighted, acknowledging that a 'one-size-fits-all' approach was unlikely to cover the complexity of need or empower individuals to engage. Building a sense of community for people living with MCN was a common suggestion aimed at reducing social exclusion. The value of peer support communities in particular was highlighted among all participants in the FGD with individuals with lived experience of MCN. The need for better collaboration across the multiple agencies involved in supporting people living with MCN was also highlighted. FGDs with staff and managers/commissioners focused more on the financial and competitive element of the current system. In particular, improved communication between services and the opportunity to share learning across sectors were suggested as ways to reduce missed opportunities for prevention.

## DISCUSSION

This is the first study, to the best of our knowledge, to use peer research approach to explore individual and service provider perspectives and experiences on factors that could contribute to high mortality rates among individuals living with MCN at the local level. Power within co-produced research can manifest in a number of ways and can result in the focus being placed on capacity building versus reflecting the topic aspirations of peer researchers.[27 28] In contrast, our study topic was solely determined by peer researchers, thereby reflecting their aspirations, while providing an opportunity to empower through capacity building as previously suggested. This

**Table 3** Key quotations illustrating subthemes within understanding factors underlying premature mortality in MCN

| Theme | Quotes |
|---|---|
| Burden of mental ill health and substance misuse issues | "Most of the people I know that's died, their mental health has just been shot to bits, it's all about the drugs. They're taking the drugs because of mental health, is that bad? I'd say more the mental health killed them…the drugs just done that job." —Individual with lived experience of MCN |
| | "These [new psychoactive substances] are completely changing the conversation to what they were 10, 15 years ago because these drugs, how they work, how quickly they hit, how quickly they can be produced, how quickly for many of them you're on cloud 9, you're away from it, 15 minutes later you're back as a normal person. Within those 15 minutes what damage you could have done to yourself, to your life, to other people, to other people's lives." —Individual with lived experience of MCN |
| | "And there's a reason why people are taking spice, because oblivion is better than reality. That's the truth of it. It's a much better option facing up to what society is." —Individual with lived experience of MCN |
| | "I just find that people who have got mental-health issues and also have addiction problems fall through the gaps, time and time again." —Front-line staff |
| | "The waiting list for CAMHS [child and adolescent mental health services] is ridiculous. You've got to be well up there on the scale to get referred. Someone with a little bit of anxiety is not going to get put through to CAMHS, whereas that anxiety will then just carry on getting worse and worse and worse, and then you end up with someone with real mental-health issues." —Front-line staff |
| Lack of hope, stigma and health-seeking behaviour | "You don't see another way…it's just doom and gloom and like you say this one's dead, this one's in prison, there's nothing ever…it's like being in the sort of devil's dungeon, to be honest." —Individual with lived experience of MCN |
| | "This is my life, so it's not a care in the world if it is death. Death has got to be better. So to be honest living that life, to some I think death would just be a much easier answer." —Individual with lived experience of MCN |
| | "It's a remembrance wall. So all the people in the service that have passed away and every time you go in it's just getting more, you have to squeeze the names in of the people that have been lost and I know them all, I'm going, 'Oh so and so has died.'" —Individual with lived experience of MCN |
| | "'Oh you're an addict,' the look on their face that it changes visibly, they treat you completely differently, 'Oh he knows.'" —Individual with lived experience of MCN |
| | "I think that word as well, like, junky really boils my blood, heroin addict is much better. Just picking up the junky, junky, junky, that's all we get." —Individual with lived experience of MCN |
| | "…this doctor at the time of [the] appointment isn't going to be able to comprehend even a tiny touch of what your life is." —Individual with lived experience of MCN |
| | "We talked to the young people who have got mental health issues and they're kind of like, 'Oh no, I don't want to talk to anyone, I don't want to tell them I've got a problem. I'd rather just smoke some weed or take some grass and I'll be okay.'" —Manager/Commissioner |
| Poor navigation and limited services | "My staff are supposed to spend their time navigating and signposting and supporting people into other services, probably about 60% of their time is now spent doing benefits stuff, just so that people have got enough money in their pockets." —Manager/Commissioner |
| | "I'm facing this maze full of doors and every time I open a door, there's another door, sets of doors. There's no coherent structure within the system that says, Here's a person who is asking for help, who's engaging with everything that we're giving, can we please pull this together so we can actually provide the help that this person needs." —Individual with lived experience of MCN |
| | "There are no youth services left…there's nothing left, and that was a huge safety net. It was a learning experience, it was preventive, and it was a place of safety for youth, and it's not there anymore." —Manager/Commissioner |
| | "It's often such a desperate situation that we're having ridiculous conversations that we want someone to be sectioned or we want someone to go to prison just so they're in some kind of contained environment where we feel we can try and manage some of the risks." —Manager/Commissioner |

Continued

| Table 3 | Continued |
|---------|-----------|
| **Theme** | **Quotes** |
| CAMHS, child and adolescent mental health services; MCN, multiple complex needs. | |

study used FGDs to gather insights and perspectives from individuals with lived experience, front-line workers and those responsible for managing and commissioning services with regard to the high mortality rates among MCN populations within the region and opportunities for potential interventions. Across discussions, we found that issues relating to the burden of mental ill health and substance misuse (dual diagnosis), combined with experiences of stigma and exclusion when accessing services, were perceived as informing premature mortality in this group. Participants highlighted key life events (eg, childhood adversity) and transitions (eg, release from prison) as potential opportunities for targeted intervention to better support this population in the future.

This study provides valuable qualitative insights both from people living with MCN and those delivering and commissioning services on the perspectives and experiences on how co-occurring mental ill health and substance misuse might combine to affect health outcomes and may lead to premature mortality. Fitzpatrick[29] theorises that causes of homelessness occur on at least four levels—economic, housing, interpersonal and individual—with no one level considered greater than another. The interlocking nature of these causes results in individuals having different experiences of homelessness and some (those who experience MCN) requiring more services and support than others. As others have reported,[18] we found that the reality of many individuals with MCN is a journey dominated by the challenge of navigating a siloed, highly fragmented system that is ill-equipped to meet their needs. Not unique to this study are the experiences of many people with MCN finding the co-occurring nature of issues leads to an inability to access services or falling through system 'cracks'.[19] The

| Table 4 | Key quotations illustrating themes within opportunities for intervention to prevent mortality in MCN groups |
|---------|-----------|
| **Theme** | **Quotes** |
| Intervention timing | "Preventive measures early on may stop the numbers of people coming through with multiple and complex needs. So it's the preventive, it's the community centres, it's the youth centres, it's those things where the learning happens." —Manager/Commissioner |
| | "People leaving prisons, we know the times when you've been drug free and then you transition in and out of prison etc. or transition from being clean for a bit to then lapsing, acute risk there. That would be a flag." —Individual with lived experience of MCN |
| | "We all know times of peak vulnerability, they don't need to be necessarily shared emotionally…people leaving prison…loss, bereavement, grief, divorce." —Individual with lived experience of MCN |
| | "There's often an inability to exploit windows of opportunity where…support workers will try and get all their ducks in a row. So the mental health stuff, the mental health treatment, housing, benefits, all of that sort of stuff, it's rare that you're going to manage to get all of that sorted in the two hours of window opportunity you've got. Then the ship sails sometimes and you don't know whether that's going to come back again or when it's going to come back again." —Manager/Commissioner |
| Intervention approaches | "I think there needs to be focus on it being really a person-centred approach and say, 'This isn't working for me at the moment and that's how I would like things to be,' and giving them that sense of responsibility." —Front-line staff |
| | "There's a connection [as a peer supporter]. Immediately there's a connection but through that connection you feel like you've gotten in and you feel like what you say has a better chance of making a difference to that person." —Individual with lived experience of MCN |
| | "I volunteer in services and in statutory services and the differences you see and people's attitudes towards you when they learn that you're an addict or an ex-addict, I'm still an addict. I am an addict, I just don't use. The difference that you see in people's faces when they say, 'Oh you're an addict,' the look on their face that it changes visibly, they treat you completely differently, 'Oh he knows.'" —Individual with lived experience of MCN |
| | "We exist in a competitive tendering landscape and we need to leave that aside and come together and share good practice and learn from what's happening across the world." —Manager/Commissioner |
| | "We need as people for services to be talking to one another to be sharing our data, to be aware of all of the needs because…that's how we get rounded people by having well rounded service provision." —Individual with lived experience of MCN |
| MCN, multiple complex needs. | |

experience of facing endless closed doors and no infrastructure in place to provide support irrespective of the access point reiterates the system not being designed for the multilevel adversity MCN populations experience.

Stigma is a well-described phenomenon in this population.[5] [8] [18] Our study highlights that stigma and the associated discrimination experienced by people with MCN also acts as a barrier to accessing support and was perceived as contributing to the high rates of mortality in this population. The perspective on experiences of threat shared by individuals with lived experience regarding the impact of stigma on health-seeking behaviours and the impact of negative experiences using acute services may inform subsequent health behaviour and may contribute to decisions leading to premature mortality. If an individual believes they will be dismissed or marginalised by a provider, there is the potential that this may increase the likelihood that they do not pursue services when needed. This notion of group marginalisation may be part of the premature mortality problem, and our qualitative study begins to capture and articulate this.

Experiences across FGDs highlight the struggle within the system to meet current demands and have positive interactions and outcomes for individuals with MCN. This is consistent with current literature.[5] Participants in the focus groups pointed to issues with the present system and identified that early prevention and targeting interventions at 'critical life events' (ie, transitions in service provision or at experiences of heightened adversity) could be important in reducing deaths in MCN groups. Through connecting individuals with support during these 'critical life events', it increases their ability to bounce back and develop resiliency. There was a particular emphasis on creating interventions targeting youth to address concerns earlier on in the prevention spectrum. Evidence highlighting the link between experiences of childhood adversity and later life MCN reiterates the need to create preventive interventions surrounding this time period.[5] [7] [30] Additionally, our findings suggest that future provision should focus on interventions that are developed collaboratively across sectors,[5] [31] targeting critical life events[7] [16] [30] using person-centred and trauma-informed practices,[31] [32] and peer support.[33]

The novelty of this study is that it has shown the value of listening to and conducting research with individuals with experience of MCN. It enabled an exploration of an issue that directly affects their community and supported understanding of the personal perspectives of a handful of people with MCN and their carers on what may have informed premature mortality among their networks, as well as identifying perceived avenues for possible preventive interventions within the local region. This enrichment of the research aligns with existing literature exploring peer research use for vulnerable populations or sensitive topics.[8] [20–25] Furthermore, the insights specific to opportunities for service provision take into consideration the lived experience of the target population, which can lead to more equitable service delivery and engagement.[21]

## Limitations

As our study was located in the North East of England, our findings may not be generalisable to other regions. The study collected data through three FGDs which included representation from MCN groups and from different sectors of support services. Although the study sample had a broad range of representation, the findings may not be generalisable to other contexts. Nonetheless, the results of this study have provided qualitative insights into the perspectives on mortality rates among MCN populations and potential directions for future intervention research.[34] Response bias is a possibility as individuals with stronger views might have been more likely to volunteer to participate in the study. Recognising focus groups have their limitations, we provided opportunity for everyone who attended to participate and actively sought input from all participants. We feel that this reduced the potential effects as the group-based data collection allowed for a range of views to be considered and reflected on. There were limitations in the representativeness of individuals with MCN, a recruitment challenge experienced in previous studies.[21] Nonetheless, by engaging peer researchers throughout the research process, we feel this potential limitation was reduced. There is also the risk that participants might have felt more comfortable speaking to an outsider rather than a peer about certain issues.[22] However, as the study focused less on the individual and more on the overarching issues related to mortality rates, this was likely minimal.

## CONCLUSIONS

Mortality rates among individuals experiencing MCN in the UK continue to rise due to an interconnected web of disadvantage and exclusion. Findings from this study provide valuable insights from people with lived experience and staff supporting MCN groups on what may be informing premature mortality among this population. Addressing concerns related to dual diagnosis, providing support around critical life events and investing in early prevention were perceived as ways to mitigate high mortality among individuals experiencing MCN. The intersecting nature of MCN should be factored into the design of future interventions to address the challenges related to service navigation and availability. The need to support existing inequalities related to experiences of MCN and develop effective and sustainable interventions to prevent premature mortality has become even more important in light of the COVID-19 pandemic, which has exacerbated issues around service access, health inequalities and social isolation.[35]

**Acknowledgements** We would like to acknowledge the various peer researchers from Fulfilling Lives Newcastle Gateshead who contributed to all stages of this study, including developing the aim, methods, data collection and interpretation. As well, we appreciate the time and effort of all participants who offered their views on and experiences of this challenging and emotive topic.

**Contributors** Conceptualisation: JH, SER, AOD, AB, GM, EG and RP. Data acquisition: JH, AB and SER. Analysis: RP, AB, EG, GM and EAA. All authors

contributed to the interpretation of findings. All authors contributed to the writing of the paper. All authors have read and approved the final version to be submitted for consideration for publication.

**Funding** This work was funded by a small seed grant from Public Health England as part of the Research Hub Initiative. EAA, EG, GM, AOD and SER are members of Fuse, the Centre for Translational Research in Public Health (www.fuse.ac.uk). Fuse is a UK Clinical Research Collaboration (UKCRC) Public Health Research Centre of Excellence. Funding for Fuse from the British Heart Foundation, Cancer Research UK, National Institute for Health Research, Economic and Social Research Council, Medical Research Council, Health and Social Care Research and Development Office, Northern Ireland, National Institute for Social Care and Health Research (Welsh Assembly Government) and the Wellcome Trust, under the auspices of the UKCRC, is gratefully acknowledged. EAA is supported by the National Institute for Health Research (NIHR) School for Public Health Research (SPHR) Pre-doctoral Fellowship (grant reference number PD-SPH-2015). SER and AOD are members of the National Institute for Health Research (NIHR) Applied Research Collaboration (ARC) North East and North Cumbria Inequality Theme. The views expressed are those of the author(s) and not necessarily those of the NIHR, Department of Health and Social Care, Public Health England, or any of the other funding or organisational bodies.

**Competing interests** None declared.

**Patient and public involvement** Patients and/or the public were involved in the design, or conduct, or reporting, or dissemination plans of this research. Refer to the Methods section for further details.

**Patient consent for publication** Not required.

**Ethics approval** Ethical approval for the study was obtained from Newcastle University Ethics Committee (reference 11064/2018).

**Provenance and peer review** Not commissioned; externally peer reviewed.

**Data availability statement** Data are available upon reasonable request. Due to the sensitivity of the topic and the number of participants, we are not able to share focus group discussion transcripts. Summaries are available from the corresponding author on request.

**ORCID iDs**
Emma A. Adams http://orcid.org/0000-0001-7536-0658
Emma L. Giles http://orcid.org/0000-0002-2166-3300
Grant J. McGeechan http://orcid.org/0000-0002-3994-8507
Amy O'Donnell http://orcid.org/0000-0003-4071-9434

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
