## [Reviewer comments · BMJ Open]

ARTICLE DETAILS

TITLE (PROVISIONAL)	EXPLORING HIGH MORTALITY RATES AMONG PEOPLE WITH MULTIPLE AND COMPLEX NEEDS: A QUALITATIVE STUDY USING PEER RESEARCH METHODS
AUTHORS	Perry, Rachel; Adams, Emma; Harland, Jill; Broadbridge, Angela; Giles, Emma; McGeechan, Grant; O'Donnell, Amy; Ramsay, Sheena

VERSION 1 – REVIEW

REVIEWER	Alexandra Jønsson University of Copenhagen, Denmark
REVIEW RETURNED	10-Nov-2020

GENERAL COMMENTS	Review: Exploring high mortality rates among people with multiple and complex needs: a qualitative study using peer research methods. First of all, thank you for the opportunity to review this article. The aim of the study is important, and the use of peer research methods is much needed in the field. However, the article is not ready for scientific publication in its present form. Please apologize the language in this review, I am not a native speaker. This is a qualitative study building on three focus groups. That – in itself – is not much, and the major problem of the article is that the results, discussion and conclusion is over interpreting the data. In answering the research question of exploring perspectives of people with lived experiences of MCN and professionals, FGs are well-suited, however, the data is sparse and compared to for instance long-term ethnographic fieldworks addressing the same question, it is just not that convincing. In answering the second aim of obtaining insights into opportunities for preventing premature deaths etc. FGs are not adequate, or should at least not stand alone. Adding knowledge to this rather epidemiological inquiry, a systematic literature review would be better. For comparison: the SOFIA study in Copenhagen are about to conduct a pilotstudy RCT on an intervention that aims to reduce increased mortality and stigma among people with severe mental illness, including people with double-diagnosis. The intervention is co-designed with patients, relatives and health care professionals, and is based on two years intensive data collection through fieldwork among patients, relatives, professionals, general practitioners, more than 100 hours of interviews, more than 20 focus groups, an extensive literature review, several PhD and postdoc studies etc. INTRODUCTION
--

The introduction is not sufficient. It is said that “gaps remain...” p.4 l. 28, citing just one source: a methods book from 2013. The authors need to do a more extensive literature research.

A vast amount of the discussed literature in the discussion section should be presented in the introduction. Which leads back to the results – what does this study add that we did not know already? All of this epidemiological etc. literature should be used in framing the problem and research question, not for verifying findings.

METHODS

The method section is good and adequate. However, it would be great with more insight into HOW exactly the peers did contribute – not in general terms, but actual examples – that would really enrich the field of peer research methods. It is stated once (p 5) that peers suggested the use of focus groups for creating a supportive environment. But was it really necessary with peers to gain that insight? It is well-established from the psychological, social science and psychiatric literature, that people with lived experience of MCN may be in a fragile and vulnerable position, and thus choosing FG is absolute basic methodological knowledge expected from a researcher. I am convinced that the use of peers added great value, but as I said, it would be appreciated if it was more explicitly described.

The FG guide address relevant questions well-suited for discussions, and I respect that we cannot get a full transcript but a basic table on participant characteristics in all three groups is needed.

The results presented are not detailed enough. For instance, we do not get any wiser of being told that the severe burden of mental illness is felt to make substantial contributions to premature mortality etc. and then one quote. Results need to be much more detailed and contain many more quotes – did all agree on this one citation? Or did they have different views? Etc.

Another example – stigma – a well-described phenomena in this group is not explained or adding to a deeper understanding by one quote that says something about wording.

All presented results are already known from international literature, like stigma, burden of disease,. The results does not present with FGs DID contribute with – a particular insight into how ie people with lived experience with MCN address and discuss issues in a group – the wording, the conflicts, the agreements etc.

DISCUSSION

This is a qualitative study and the discussion ought to add a theoretical perspective to the results. This is not the case.

The discussion also concludes that the novelty of this study (?) have shown the value of listening to individuals with experience of MCN.

First, so many researchers in the field of ie anthropology and psychology have already done that – what does it add to existing work? And second, we have not really seen any value of listening to the individuals, as there are no results presented which we did not already know from existing literature.

LIMITATION

Three focus groups with three different groups of participants are a major limitation, and not much can truly be concluded on this basis. It should be addressed.

The choice of focus groups does not limit the participation bias – contrary, in focus groups, more quiet and introvert participants may easily have trouble getting their views voiced.

	CONCLUSION As already stated, the authors should present discussion and conclusion with much more caution.
--	--

REVIEWER	Scott Landes Syracuse University, US
REVIEW RETURNED	24-Nov-2020

GENERAL COMMENTS	This study utilized data from focus groups to better understand the perspectives of people with MCN and those who provide their services. The methods are apropos to the research question and described in sufficient detail. I have no concerns with the methods or results. My only concern is with the way in which the study is framed and findings are interpreted. Language utilized at the beginning of the paper hints at causality: “reasons underlying high mortality rates”; “significant factors underlying premature mortality.” Also, at the end of the paper, the statement is made about the need to “address the underlying causes of MCN . . . and high mortality rates.” In my mind, this language is not warranted by the data. The data utilized in the study provides the perspective of people with MCN and those that provide their services on what may be informing premature mortality. As such, these are individual opinions, and are not sufficient empirical evidence to make statements about underlying factors or causes for population health outcome. The authors appear to recognize this point in the results section, when they say on multiple occasions that respondents “felt” that X or Y was an influential factor on premature mortality. This strategy should be utilized throughout the paper – to make explicitly clear that the are the perceptions of people with MCN and their care support on what may be informing premature mortality among this population. Beyond the fact that the data analyzed constitutes personal perspectives, the very small sample size also necessitates that due emphasis is laid on the “may” in these statements. Results really provide insight on the personal perspective of a handful of people with MCN and their carers, within a particular location, on what may have informed premature mortality among their social network. That being said, even if carefully articulated as suggested, the findings are informative. Yet, further discussion could be devoted to the ways in which perspective of threat, which is what participants are expressing, may inform subsequent behavior and also contribute to decisions leading to premature mortality – if an individual thinks they will be marginalized or dismissed by a health care provider, this may increase the likelihood that they do not pursue services when needed. I really think a necessary move away from suggesting causality affords more time for the authors to discuss how perceptions of group marginalization may also be part of the problem, and one that qualitative studies such as this one have a better chance of capturing and articulating.
---

VERSION 1 – AUTHOR RESPONSE

Reviewer: 1		
Major comments: General		
1. a) This is a qualitative study building on three focus groups. That – in itself – is not much, and the major problem of the article is that	Thank you for this comment. 1.a) We agree and appreciate the need to exercise caution in interpreting the data and	Abstract and article summary first bullet (pg. 2)  Clarifying that the study focuses on the perspectives of people with MCN and those that provide their services of

the results, discussion and conclusion is over interpreting the data. 1 b) In answering the research question of exploring perspectives of people with lived experiences of MCN and professionals, FGs are well-suited, however, the data is sparse and compared to for instance long-term ethnographic fieldworks addressing the same question, it is just not that convincing. In answering the second aim of obtaining insights into opportunities for preventing premature deaths etc. FGs are not adequate, or should at least not stand alone. Adding knowledge to this rather epidemiological inquiry, a systematic literature review would be better. For comparison: the SOFIA study in Copenhagen are about to conduct a pilotstudy RCT on an intervention that aims to reduce increased mortality and stigma among people with severe mental illness, including people with double-diagnosis. The intervention is co-designed with patients, relatives and health care professionals, and is based on two years intensive data collection through fieldwork among patients, relatives, professionals, general practitioners, more than 100 hours of interviews, more than 20 focus groups, an extensive literature review, several PhD and postdoc	findings of our study. Our findings are based on focus groups held with people experiencing multiple complex needs (MCN), and those supporting them – our findings are based on interpretation of these data and perspectives only. We, have, therefore, revised the Abstract, Results, Discussions and Conclusions to emphasise that the findings are perceptions of study participants – we have made changes to ensure that there is no indication of over-interpretation of data. 1.b) In our study the objective was to gain insights from the perspectives of MCN groups and those supporting them, so as to identify opportunities for preventing high mortality rates in MCN groups. As the Reviewer points out, ours is an observational study and not an intervention evaluation or systematic review – nonetheless, the study provides important insights that are needed for developing interventions that are sustainable and relevant for local populations and systems. As the Reviewer suggests, intervention evaluation studies (for example, the SOFIA study highlighted by the Reviewer) are more suited to adopt longer-term follow-up methods with co-designing interventions. We recognise that some of the previous wording in the Introduction may have implied that our study was looking at an intervention. We have, therefore, removed that specific reference and have clarified that our study seeks to data on perceptions and	living with high premature mortality.  • Abstract objective: ‘To explore the perceived reasons underlying high mortality rates among people with multiple and complex needs’ • Article summary first bullet: ‘explore factors perceived to contribute to high mortality rates ‘ Introduction (pg. 4) Clarified in Introduction that this study explores perspectives and experiences of MCN groups and those supporting them in order to add evidence to develop future interventions (page 4).  • Adjusted language ‘ ... understand factors perceived to contribute to high mortality rates in this population’ • Added ‘This study uses peer research methods to obtain perspectives on premature mortality...’ We have clarified in the Results, Discussion and Conclusions that the findings are based on perceptions and experiences of people experiencing MCN and those supporting them – we have altered wording to ensure that we do not imply causality or over-interpret the results (page 8, 10, 11, and 12). Results (pg. 8)  • New sentence: ‘Participants in the individual with lived experience FGD shared experiences and perceptions of drugs being used as a way for individuals to escape from their day to circumstances.’ (paragraph 1) • Adjusted language ‘... large role it was perceived to play in health-seeking behaviour.’ (paragraph 4) Discussion Pg. 10 paragraph 1  • Adjusted language ‘... on perspectives into the high mortality rates...’ • Adjusted language ‘... were perceived as informing premature mortality...’ Pg. 11  • Adjusted language ‘on perspectives and experiences on how co-occurring mental ill-health and substance misuse
--	--	---

studies etc.	experiences of MCN groups – these findings will add to the evidence contributing to developing and evaluating future intervention studies. We have also clarified in the Introduction that this study provides insights from people experiencing MCN and those supporting them while utilising a peer research approach to the data collection. We have further revised the Introduction to expand on the background to the study and the peer research methods.	might combine to affect health outcomes and may lead to premature mortality.’ (paragraph 1)  Adjusted language ‘... understanding of the personal perspective of a handful of people with MCN and their carers on what may have informed premature mortality among their networks, as well as identifying perceived avenues...’ (paragraph 4) Conclusion (pg. 12)  Adjusted language ‘...people with lived experience and staff supporting MCN groups on what may be informing premature mortality among this population.’ Adjusted language ‘... early prevention were perceived as ways to mitigate high mortality rates among individuals experiencing MCN.’ Limitations (pg. 12) We have added a note in our limitations about using three focus groups in the Limitation section.  Added ‘The study collected data through three FGDs which included representation from MCN groups and from different sectors of support services. Although the study sample had a broad range of representation, the findings many not be generalizable to other contexts. Nonetheless, the results of this study have provided qualitative insight into perspectives on mortality rates among MCN populations and potential directions for future intervention research.’
Major comments: Introduction		
2a. The introduction is not sufficient. It is said that “gaps remain...” p.4 l. 28, citing just one source: a methods book from 2013. The authors need to do a more extensive literature research.	Thank you for highlighting this issue. As suggested, we have added further to the Introduction section to expand on the literature and articulate the evidence gaps. As per your suggestion in comment 2b below, we have moved some of the literature previously in the Discussion into the	Introduction pg. 3-4 (see changes for comment 2b as well) Pg. 3 Paragraph 2  Added ‘The connection between homelessness and premature mortality has been highlighted through a number of quantitative studies.(4, 10-12, 15-17)’ Added ‘However, most of the research into mortality within this population has been

	Introduction section. We believe your suggestions in this comment along with comment 2b have greatly improved the Introduction. Below are a few highlights of how it has been improved. Paragraph 2 focusing on ‘mortality’: Regarding the link with mortality we have moved the literature sentences from the discussion as well as adding an additional sentence emphasising that the connection between homelessness and mortality has been evaluated quantitatively. We have further added a final sentence to this paragraph re-iterating that the gap is in qualitative research. Paragraph 3 focusing on ‘services and perspectives of staff and individuals with lived experience’: We have reframed the sentence referring to gaps remain and believe this reframing is now situated within a larger literature base. As mentioned in response to comment 3b below, we have brought forth the references in the discussion related to the lack of capacity in the system and the fragmented/siloed nature of the system. We have also added in a final ending sentence to this paragraph highlighting the pressing need for research involving both populations. Paragraph 4 focusing on ‘peer research’: We have added an additional comment on peer research articulating the benefit of co-producing and doing research ‘with individuals with lived	epidemiological in nature and there is a gap in qualitative research.’ Pg. 3 Paragraph 3  • Reframed ‘Despite the breath of quantitative and epidemiological data on MCN and mortality, there are very limited perspectives of individuals with lived experienced of MCN and those who support them (e.g. health and social care workers) on the high death rates in this population.’ • Added the following: ‘With a system that is not designed to meet the needs of MCN populations, there is a pressing need to conduct research exploring the perspectives of both service users and providers to develop a more comprehensive picture of current challenges and potential opportunities for improvements.’ Pgs. 3-4 paragraph 4  • Added: ‘Conducting research on MCN populations can be extremely challenging; however, conducting studies with individuals with lived experience of MCN has been found to add immense value through enhancing recruitment efforts and the collective understanding of the experiences.(23-25)’
--	---	---

	experience of MCN.’	
2b. A vast amount of the discussed literature in the discussion section should be presented in the introduction. Which leads back to the results – what does this study add that we did not know already? All of this epidemiological etc. literature should be used in framing the problem and research question, not for verifying findings.	Thank you for this helpful comment. As suggested, we have moved a few of the literature and epidemiological information from the Discussion into the Introduction. For example, we have now moved the information on the higher prevalence of alcohol use and mental ill-health to immediately follow our definition of multiple complex needs. We have followed this with the sentence on epidemiological data suggests that co-occurring mental ill-health and substance misuse is high for mortality rates among people experiencing MCN. We believe this suggestion has enhanced the framing of the problem and highlighted the quantitative focus of existing mortality data. We have also moved the service mapping sentence into the introduction and have referenced the two related references presented in the discussion in this section of the introduction.	Introduction (pg.3) Paragraph 2 Moved the following from the Discussion into the Introduction section:  • ‘Homeless individuals have greater prevalence of alcohol use disorders than the general population,(9) and self-reported data suggest that mental health problems are as high as 92% among individuals with MCN.(5)’ • ‘Epidemiological data suggests that the contribution of mental ill-health and co-occurring substance misuse is high for mortality rates among people experiencing MCN.(10, 11)’ Paragraph 3 Moved the following from the Discussion into the Introduction  • ‘A recent mapping exercise of MCN across England found the current system of support across public and voluntary services was struggling to meet present demands and deliver positive outcomes for individuals with MCN.(5)’ Added the following based on literature presented in the discussion  • This is increasingly problematic when the system is fragmented and siloed causing added barriers to much needed support. (18, 19)
Methods		
3. The method section is good and adequate. However, it would be great with more insight into HOW exactly the peers did contribute – not in general terms, but actual examples – that would really enrich the field of peer research methods. It is stated once (p 5) that peers suggested the use of focus groups for creating a supportive environment. But was it really necessary with peers to gain that insight? It is well-	Thank you for this feedback. We have further refined the Methods section and added information to be more explicit about the contribution of peer researchers. Peer researchers contributed to all aspects of this study, starting from defining the research question, informing data collection methods, assisting with data collection, and interpretation of findings. We accept the need to make all this more explicit in our paper. We have added this added	Methods paragraph 1 (pg. 4) We have added the following information to make clearer the role and contributions of peer researchers in this study:  • ‘Peer researchers identified the study topic as an area that was personally impacting them and required attention. Additionally, peer researchers determined the data collection approach (FGDs), co-developed the topic guide, assisted facilitating focus groups, contextualised the findings, selected quotes, presented findings at a dissemination event, and co-authored a blog on the topic.’

establish from the psychological, social science and psychiatric literature, that people with lived experience of MCN may be in a fragile and vulnerable position, and thus choosing FG is absolute basic methodological knowledge expected from a researcher. I am convinced that the use of peers added great value, but as I said, it would be appreciated if it was more explicitly described.	information now in the Methods section.	
4. The FG guide address relevant questions well-suited for discussions, and I respect that we cannot get a full transcript but a basic table on participant characteristics in all three groups is needed.	Thank you for this comment we have included the available participant characteristics within the paper. At the strong recommendation of the peer researchers, we only collected gender information for the FGD for individuals with lived experience. For the FGDs with staff and managers/commissioners, we focused on recruiting based on diversity of service provision. As such, we have included information to reflect this.	Methods data collection section (pg. 5)  • Inserted a participant characteristic table based on available data (see Table 1)
Major comments: Results		
5a. The results presented are not detailed enough. For instance, we do not get any wiser of being told that the severe burden of mental illness is felt to make substantial contributions to premature mortality etc. and then one quote. Results need to be much more detailed and contain many more quotes – did all agree on this one citation? Or did they have different views? Etc.	Thank you for pointing out this concern. We had previously presented quotes that were most in agreement, but have moved quotes from within the Discussion and have instead presented a collection of quotes (both previously presented and additional ones) within newly introduced Tables 3 and 4. We have included language to reflect agreement and differences for themes.	Results (pgs. 6-10) We have removed quotes from within subsections of the results and have instead now created two tables – Tables 3 (pgs. 6-8) and 4 (pgs. 8-9) where we present the previous quotes alongside additional quotations. We have adjusted and included language to reflect agreement and differences for themes:  • Added: 'Quotes most in agreement are presented in Tables 3 and 4. Quotes from individuals with lived experience of MCN were more likely to focus on their personal experience as opposed to system factors as identified by staff groups. There was a high level of agreement in data collection both within and across the focus groups.' (pg.

		6) Burden of mental ill-health and substance misuse issues (pg. 8)  • Added text: 'Participants in the individual with lived experience FGD shared experiences and perceptions of drugs being used as a way for individuals to escape from their day to day circumstances.' (paragraph 1) • Adjusted language: 'Across all three FGDs, there was a perception...' (paragraph 2) Lack of hope, stigma, and health-seeking behaviour (pg. 8)  • Added: 'Stigma was discussed within the individual with lived experience FGD in relation to the large role...' (paragraph 2) Intervention approaches (pg. 10)  • Added: 'Across all FGDs, participants provided...' • Added: '... highlighted among all participants in the FGD with individuals with lived experience of MCN.' • Added: 'FGDs with staff and managers/commissioners focused more on the financial and competitive element of the current system.'
5b. Another example – stigma – a well-described phenomena in this group is not explained or adding to a deeper understanding by one quote that says something about wording.	Thank you for pointing out this concern. In the context of this study, stigma arose as an issue that emerged from the data collected. Stigma was referred to by participants in relation to the role it plays in health seeking behaviour, particularly as a barrier to accessing treatment and services. We have added an additional quote on this point. We have also clarified that stigma was discussed in relation to health-seeking behaviour. Additionally, we have included additional information in the Discussion section, to clarify that the issue of stigma is in the context of a barrier to health-seeking behaviour. Further detail is provided in relation to Reviewer 2's comment 3.	Results (pgs. 7-8) Table 3 (pg. 7)  • Added stigma related quotation: 'Oh you're an addict,' the look on their face that it changes visibly, they treat you completely differently, 'Oh he knows.' – individual with lived experience of MCN Stigma paragraph (pg. 8)  • Added text to further clarify the point of stigma as expressed by participants: 'Stigma was discussed within FGDs in relation to the large role it was perceived to play.... around the facial expressions and stigmatising nature...'

5c. All presented results are already known from international literature, like stigma, burden of disease,.The results does not present with FGs DID contribute with – a particular insight into how ie people with lived experience with MCN address and discuss issues in a group – the wording, the conflicts, the agreements etc.	Thank you for this comment. We agree and have provided greater clarity and information on how MCN groups perceive issues related to high mortality rates. The results presented are based on themes that emerged from the data – these themes are based on perceptions and experience of people experiencing MCN, or those working to support them. As the Reviewer points out, some issues that emerged (for example, stigma, dual diagnosis) have been previously reported in the literature. It is interesting that these issues were highlighted as being perceived to be factors that also underlie high mortality rates. We have now added this to the Results and Discussion sections. The results were based on agreements both within and between the focus groups, which we have also clarified in the Results.	We have added further text in Results (page 6) and Discussion (page 11). Results As presented in response to comment 5a, we have added additional language to emphasise when findings were in agreement. Pg. 6  Added the following sentence before presenting the quotes: 'There was a high level of agreement in data collection both within and across the focus groups.' Pg. 8  Added the following sentence to emphasise a finding unique to the FGD with individuals with lived experience of MCN: 'Participants in the individual with lived experience FGD shared experiences and perceptions of drugs being used as a way for individuals to escape from their day to day circumstances.' Adjusted language: 'Stigma was discussed within the individual with lived experience FGD in relation to the large role it was perceived to play in health-seeking behaviour.' Pg. 10  Adjusted language 'The value of peer support communities in particular was highlighted among all participants in the FGD with individuals with lived experience of MCN.' Added: 'FGDs with staff and managers/commissioners focused more on the financial and competitive element of the current system.' Discussion (pg. 11) Dual diagnosis and burden of disease (discussion paragraph 2)  Adjusted language: 'This study provides valuable qualitative insights from both people living with MCN and those delivering and commissioning services on perspectives and experiences on how co-occurring mental ill-health and substance misuse might combine to affect health outcomes and may lead to
--	--	--

		premature mortality.’  • Added: ‘The interlocking nature of these causes result in individuals having different experiences of homelessness and some (those experiences MCN) requiring more services and support than others.’ • Added: ‘The experience of facing endless closed doors and no infrastructure in place to provide support irrespective of the access point reiterates the system not being designed for the multi-level adversity MCN populations experience.’ Stigma (discussion paragraph 3) We have incorporated the suggestion made by Reviewer 2 in their comment 3 to highlight the unique role of stigma within this study. We have greatly revised this point in the discussion and it has now become an entire paragraph.  • Adjusted language: ‘Our study highlights that stigma and the associated discrimination experienced by people with MCN also acts as a barrier to accessing support and was perceived as contributing to the high rates of mortality in this population.’ • Added: ‘This notion of group marginalisation may be part of the premature mortality problem, and our qualitative study begins to capture and articulate this.’
Major comments: Discussion		
6a This is a qualitative study and the discussion ought to add a theoretical perspective to the results. This is not the case.	Thank you for this helpful comment. We agree and have added more information on theoretical perspectives to contextualise and understand our findings. Our study was strongly influenced by its co-production with peer researchers and took more of a deductive approach. We have referenced Guta, Flicker, and Roche (2013) along with Foucault in our discussion in relation to the role of peer researchers and power. Further, we have referenced Fitzpatrick (2005) for theories related to the different causes of	Discussion (pgs. 10-11):  • Added: ‘Power within co-produced research can manifest in a number of ways and can result in the focus being placed on capacity building versus reflecting the topic aspirations of peer researchers.(27, 28) In contrast, our study topic was solely determined by peer researchers thereby reflecting their aspirations while providing an opportunity to empower through capacity building as suggested in previous research.’ (paragraph 1 in discussion) • Added: ‘Fitzpatrick theorises that causes of homelessness occur on at least four levels—

	homelessness in relation to the wider system required to support those experiencing MCN; this conceptual framework is also apparent in our study findings which points to the role of multiple factors underlying high mortality in MCN groups.	economic, housing, interpersonal, and individual—with no one level considered greater than another.(29) The interlocking nature of these causes result in individuals having different experiences of homelessness and some (those experiences MCN) will require more services and support than others.’ (paragraph 2 in discussion)
6b the value of listening to individuals with experience of MCN. First, so many researchers in the field of ie anthropology and psychology have already done that – what does it add to existing work? And second, we have not really seen any value of listening to the individuals, as there are no results presented which we did not already know from existing literature.	Thank you for your comment. We agree that fields such as anthropology and psychology draw quite a lot on insights of individuals. However, within current published public health research on mortality among MCN populations, studies have mostly presented quantitative and epidemiological causes of mortality in MCN groups. To our knowledge, our is the first study that has involved peer researchers to explore the perspectives of both service providers and MCN groups to understand their perspectives and experiences in relation high mortality rates in MCN populations. We agree that we need to make this clearer and have now added further text on this point in the Introduction and the Results sections. We also agree there is a need to add clarity on the added value of including the perspectives of people experiencing MCN and also of the use of peer research methods – we have now done so in the Introduction, Methods and Discussion sections (as described above in response to comment 3).	Introduction (pgs. 3-4)  Added: ‘Conducting research on MCN populations can be extremely challenging; however, conducting studies with individuals with lived experience of MCN has been found to add immense value through enhancing recruitment efforts and the collective understanding of the experiences.(23-25)’ Methods (pg. 4)  Added: ‘Peer researchers identified the study topic as an area that was personally impacting them and required attention. Additionally, peer researchers determined the data collection approach (FGDs), co-developed the topic guide, assisted facilitating focus groups, contextualised the findings, selected quotes, presented findings at a dissemination event, and co-authored a blog on the topic.’ Discussion (pg. 10)  Added: ‘This is the first study to the best of our knowledge which uses peer research approaches to explore individual and service provider perspectives and experiences of high mortality rates for individual living with MCN at a local level.’

Major comments: Limitation		
7. Three focus groups with three different groups of participants are a major limitation, and not much can truly be concluded on this basis. It should be addressed.	We thank the Reviewer for pointing this out. We undertook data collection in the form of focus groups as this was deemed to be most appropriate and sensitive by our peer researchers to discuss the issue of mortality. As with any research methodology, there are pros and cons with focus groups and interviews. On balance, we believe that the breadth and depth of data collected through our focus groups do not point to any limits in the insights and information that our study aimed to gather. We were also able to have representation from MCN groups and from different support service sectors. We were able to glean data on different aspects of our study objectives. Having focus groups allowed representation from a variety of different people (from individuals with lived experience to different providers). As the Reviewer suggested, we have included further detail on participants information to provides a better understanding of the participant pool. Furthermore, we have addressed potential limitations associated with focus groups in the Discussion section. As with any such study, generalisability can be a potential limitation. Nonetheless, we were exploring the experiences of our participants in their regional context, while strengthening this through including to both individuals with lived experience of MCN and those who support them in the study sample. We have also included a reference to an	Limitations section (pg. 12)  Added: 'The study collected data through three FGDs which included representation from MCN groups and from different sectors of support services. Although the study sample had a broad range of representation, the findings many not be generalizable to other contexts. Nonetheless, the results of this study have provided qualitative insight into perspectives on mortality rates among MCN populations and potential directions for future intervention research.(34)'

	article which highlights the use of qualitative studies in providing initial insights into perspectives, which allow for larger future studies.	
8. The choice of focus groups does not limit the participation bias – contrary, in focus groups, more quiet and introvert participants may easily have trouble getting their views voiced.	Thank you for this comment. We appreciate that focus groups have the potential limitation that quieter participants may not be confident to share their opinions. In our focus groups, the facilitators ensured that all participants had opportunities to express themselves. Having a peer researcher co-facilitate the focus group also helped with this sensitivity. We have added this clarification in the Methods and Discussion sections.	Limitations section (pg. 11)  Changes: ‘Recognising focus groups have their limitations, we provided opportunity for everyone who attended to participate and actively sought input from all participants. We feel that this reduced the potential effects as the group-based data collection allowed for a range of views to be considered and reflected on.’
Major comments: Conclusion		
9. As already stated, the authors should present discussion and conclusion with much more caution.	Thank you for raising this concern and echoing the importance of being more cautious with our language in the conclusion. As described in our response to comment 1a above, we agree and accept the need to be cautious in interpreting our findings.	See response to comment 1 above and response to Reviewer 2 comment 2 below.
Reviewer: 2		
Major comments		
This study utilized data from focus groups to better understand the perspectives of people with MCN and those who provide their services. The methods are apropos to the research question and described in sufficient detail. I have no concerns with the methods or results.	Thank for your positive and constructive feedback – as below, we have responded to your feedback and hope these revisions further strengthen our manuscript.	
1. My only concern is with the way in which the study is framed and findings are interpreted. Language utilized at the beginning of the paper hints at causality: “reasons underlying high	Thank you for raising this concern and reminding us to ensure the language does not hint at causality. We agree that we need to be cautious so as not to over-interpret the data and results. Ours is an	Abstract (pg. 2)  Adjusted language in objective: ‘To explore the perceived reasons underlying high mortality rates among people with multiple and complex needs.’ Adjusted language in the

mortality rates”; “significant factors underlying premature mortality.” Also, at the end of the paper, the statement is made about the need to “address the underlying causes of MCN . . . and high mortality rates.” In my mind, this language is not warranted by the data. The data utilized in the study provides the perspective of people with MCN and those that provide their services on what may be informing premature mortality. As such, these are individual opinions, and are not sufficient empirical evidence to make statements about underlying factors or causes for population health outcome. The authors appear to recognize this point in the results section, when they say on multiple occasions that respondents “felt” that X or Y was an influential factor on premature mortality. This strategy should be utilized throughout the paper – to make explicitly clear that there are the perceptions of people with MCN and their care support on what may be informing premature mortality among this population.	observational study based on perceptions and experiences of people from MCN groups or those supporting them. We have ensured now to clarify that the findings are perceptions of study participants. We have revised the manuscript throughout (Abstract, Article Summary, Introduction, and Discussion) to address these concerns and, in some areas, used language as per your suggestion. We were grateful for these suggestions.	results: ‘Findings from this study provide valuable perspectives of people with MCN and those that provide them with support on what may be perceived factors underlying premature mortality. Mental ill-health and substance misuse (often co-occurring dual diagnosis) were perceived as influencing premature mortality among multiple and complex needs groups. Perceptions of opportunities for identify people at-risk ...’  Adjusted language in conclusion: ‘Future interventions could take into consideration...’ Article summary (pg. 2)  Adjusted language in first bullet: ‘This study employed focus group discussions with individuals with multiple and complex needs and service providers to explore factors perceived to contribute to high premature mortality rates for individuals experiencing multiple and complex needs.’ Adjusted language in last bullet: ‘...might have led the results to being more generalizable; however, this study was focused on a local level.’ Introduction (pg. 4)  Adjusted language: ‘aim of this study was to explore the perspectives of people with lived experience of MCN and professionals who support them in order to understand factors perceived to contribute to high mortality rates in this population. This study uses peer research methods to obtain perspectives on premature mortality and rich insights into opportunities for preventing deaths and improving service provision for MCN groups in the future.’ Discussion (pgs. 10-11): Paragraph 1 pg. 10  Changes: ‘perspectives into the’ rather than ‘factors underlying’ (pg. 10 paragraph 1 in discussion)
---	---	---

		 • Changes: ‘perceived as informing premature mortality’ rather than ‘felt to contribute to adverse mortality outcomes’ • Added: ‘within the region’ • Added: ‘perceived as informing premature mortality’ Paragraph 2 pg. 11  • Adjusted language: ‘This study provides valuable qualitative insights from both people living with MCN and those delivering and commissioning services on perspectives and experiences on how co-occurring mental ill-health and substance misuse might combine to affect health outcomes and may lead to premature mortality.’ Paragraph 3 pg.11  • Adjusted language: ‘Our study highlights that stigma and the associated discrimination experienced by people with MCN also acts as a barrier to accessing support and was perceived as contributing to the high rates of mortality in this population.’ Paragraph 5 pg. 11  • Changes: ‘... understanding of the personal perspective of a handful of people with MCN and their carers on what may have informed premature mortality among their networks’ rather than ‘understanding of some of the underlying issues’ • Adjusted language: ‘... as well as identifying perceived avenues for...’ Conclusion (pg. 12):  • Adjusted language: ‘what may be informing premature mortality among this population’ rather than ‘the complexity underpinning this trend’ - Adjusted language: ‘Addressing concerns...were perceived as ways to mitigate high mortality...’ rather than ‘reducing mortality rates among MCN groups requires addressing issues’ - Adjusted language: ‘support existing inequalities related to experiences of MCN’ rather than ‘address the underlying causes of MCN’
--	--	---

		 - Adjusted language: 'prevent premature mortality' rather than 'address high mortality rates'
2. Beyond the fact that the data analyzed constitutes personal perspectives, the very small sample size also necessitates that due emphasis is laid on the "may" in these statements. Results really provide insight on the personal perspective of a handful of people with MCN and their carers, within a particular location, on what may have informed premature mortality among their social network. That being said, even if carefully articulated as suggested, the findings are informative.	Thank you for this comment and your thoughts onto the insight this study has to offer. As presented in our responses to your comment 1 above, we believe we have laid emphasis on using 'may' or 'could'. We have now checked and revised the text to ensure that we have avoided wording that implied causality in interpretation of findings. As the Reviewer suggests, our findings are informative in terms of providing perspectives of MCN groups and service providers, and these findings could be informative in developing further interventions.	Changes made as described above in comment 1
3. Yet, further discussion could be devoted to the ways in which perspective of threat, which is what participants are expressing, may inform subsequent behavior and also contribute to decisions leading to premature mortality – if an individual thinks they will be marginalized or dismissed by a health care provider, this may increase the likelihood that they do not pursue services when needed. I really think a necessary move away from suggesting causality affords more time for the authors to discuss how perceptions of group marginalization may also be part of the problem, and one that qualitative studies such as this one have a better chance of capturing and articulating.	We are grateful for this really helpful suggestion. We agree that the issue of stigma, and related marginalisation and fear of seeking support as an experience of threat, could potentially contribute to premature mortality. Stigma was a strong theme in our findings, and we agree that expanding on this further would add to the Discussion of our results. As suggested, we have included this suggestion within the paper as it enhances this Discussion paragraph.	Discussion paragraph 2 (pg. 10)  • Added 'The perspective on experiences of threat shared by individuals with lived experience, regarding the impact of stigma on health-seeking behaviours and the impact of negative experiences using acute services, may inform subsequent health behaviour and may contribute to decisions leading to premature mortality. If an individual believes they will be dismissed or marginalised by a provider, there is the potential that this may increase the likelihood that they do not pursue services when needed. This notion of group marginalisation may be part of the premature mortality problem, and our qualitative study begins to capture and articulate this.'

VERSION 2 – REVIEW

REVIEWER	Scott Landes Syracuse University, US
REVIEW RETURNED	03-Feb-2021
GENERAL COMMENTS	My concerns were addressed. Look forward to seeing this study published.